# Mental Health and Quality of Life among Dental Students during COVID-19 Pandemic: A Cross-Sectional Study

**DOI:** 10.3390/ijerph192114061

**Published:** 2022-10-28

**Authors:** Maja Milošević Marković, Milan B. Latas, Srđan Milovanović, Sanja Totić Poznanović, Miloš M. Lazarević, Milica Jakšić Karišik, Jana Đorđević, Zoran Mandinić, Svetlana Jovanović

**Affiliations:** 1Department of Public Health, School of Dental Medicine, University of Belgrade, Dr Subotica 1, 11000 Belgrade, Serbia; 2Faculty of Medicine, University of Belgrade, Dr Subotica 8, 11000 Belgrade, Serbia; 3Clinic for Psychiatry, University Clinical Center of Serbia, Pasterova 2, 11000 Belgrade, Serbia; 4Clinic for Pediatric and Preventive Dentistry, School of Dental Medicine, University of Belgrade, Dr Subotica 11, 11000 Belgrade, Serbia

**Keywords:** COVID-19, dental environment, public health, quality of life, Patient Health Questionnaire (PHQ-9), generalized anxiety disorder (GAD), dental students

## Abstract

Students are particularly vulnerable from the mental health aspect, which was especially recognized during the COVID-19 pandemic. This study aimed to reveal the impact of COVID-19 on quality of life (QoL) and mental health among dental students. The study was conducted on a sample of 797 students (207 male and 592 female) with an average age of 21.7 ± 2.4, from the School of Dental Medicine, University of Belgrade. The measurements used in the study were the Demographic and Academic Questionnaire, Questionnaire about exposure to COVID-19, COVID-19-Impact on QoL Questionnaire (COV19-QoL), Generalized Anxiety Disorder 7-item (GAD-7) scale, and Patient Health Questionnaire (PHQ-9). The mean total score for COV19-QoL was 2.9 ± 0.9, while the diagnostic criteria of GAD-7 and depression met 19.9% and 31.4% of students, respectively. There was a positive and strong correlation between QoL, anxiety, and depression. During COVID-19, predictors for lower perceptions of QoL were female gender and death of close relatives (*p* = 0.049, *p* = 0.005, respectively). At the same time, predictors for GAD were female gender, living in dormitories, and death of close relatives (*p* = 0.019, *p* = 0.011, *p* = 0.028, respectively), while for depression they were year of study, living with parents, and death of close relatives due to COVID-19 (*p* = 0.012, *p* = 0.008, *p* = 0.029, respectively). The study showed that students’ QoL and mental health during the pandemic were at high risk.

## 1. Introduction

The Coronavirus Disease 2019 (COVID-19) was defined as an extreme global health, economic, and social emergency by the World Health Organization (WHO) in March 2020 [1]. This pandemic is a unique worldwide experience in modern history. Many studies identified its tremendous impact on physical health but also mental health and quality of life in general [2,3,4,5].

The first confirmed case of COVID-19 infection in Serbia was reported on 6 March 2020 and the first COVID-19-related death was announced on 2 February. In Serbia, the lockdown has been implemented from March until May to prevent the spread of the infectious pandemic [6]. In that period, in addition to the suspension of activities at the university, all student dormitories were also closed. After reopening in May, an increasing number of students showed signs of infection by COVID-19 in several distinctive waves [7].

The University of Belgrade, as a public university located in a large metropolitan area, started using a hybrid teaching system. The university organized online lectures with practical classes held physically on site in reduced groups of students. All epidemiological recommendations given by the Ministry of Education, Science and Technological Development of Serbia were adopted to prevent the spread of COVID-19. In the following period, further enforcing social distancing, reducing working hours in stores and restaurants, gyms, sports facilities, and theaters, and limiting traveling and socializing were recommended.

Studies show that epidemiological measures such as lockdowns and social isolation can decrease the spreading of COVID-19 but can also cause psychological distress, anxiety, and depression [4,8,9]. Although young people are less exposed, some studies have shown that epidemiological measures taken to prevent the spread of COVID-19 infection had a more emotional effect on younger people than on other age groups [10,11].

Regardless of good physical health, mental health disorder symptoms were widespread in the student population, exposing students as a particularly vulnerable group in terms of mental health during the COVID-19 pandemic [7,12,13,14,15]. Reports show that medical students, and particularly dental students, even before the pandemic, present a higher level of stress during education, which considerably impacts their quality of life (QoL) [7,16,17].

Undergraduate programs in dentistry are characterized by extensive theoretical learning during the pre-clinical period, while the later period includes basic concepts for dental practice and the development of clinical skills necessary for professional activity [16,17,18]. In both cases, the transfer of knowledge is more efficient in direct contact with teachers and patients and vital in acquiring clinical skills, which was hard to do during the pandemic [19,20].

It is already known in the literature that the COVID-19 pandemic affected clinical dental education and clinical dental practice in general [21]. Furthermore, reports show that dental students are stressed due to a lack of clinical skills caused by the pandemic and worried about not becoming good enough dentists after graduation [22]. Due to the COVID-19 pandemic, there is a decrease in the quality of life among dental students who received online/distance learning [20]. Recent studies have shown that population of dental students experienced significant levels of anxiety and depression during COVID-19 [20,23,24,25,26,27,28].

Psychometric scales are one of the most often used research methods in sciences providing reliable and valid measures of mental health statuses [29]. There are widely available psychometric tools developed to study the psychosocial impact of the COVID-19 pandemic, such as COV19-impact on quality of life, GAD-7, and PHQ-9. These instruments presented good psychometric characteristics and quality in the general population [29,30,31].

The aims of the present study were to (1) investigate the impact of the COVID-19 pandemic on quality of life and mental health (anxiety and depression) and (2) identify significant predictors of the quality of life, levels of anxiety, and depression in a sample of dental students by analyzing a number of demographic and academic characteristics and their exposure to COVID-19.

The findings identify a vulnerable subpopulation of dental students who should receive special attention in order to preserve and improve their quality of life and mental health.

## 2. Materials and Methods

The study was approved by the Ethical Committee of the School of Dental Medicine, University of Belgrade (No. 36/4) and conducted in accordance with the Declaration of Helsinki.

### 2.1. Participants

A cross-sectional study was conducted at the beginning of the winter semester in 2021 (4–8 October) during the mandatory introductory practical classes in classrooms at the School of the Dental Medicine University of Belgrade, Serbia. The students were selected in the order of appearance regardless of the year of study. Participation was anonymous and voluntary. All students provided written informed consent to participate in this study.

The sample consisted of 867 students. Forty-three students refused to participate in the study, while twenty-seven provided invalid data. The final sample included 797 students, male (n = 207, 26%) and female (n = 592, 74%) with average age 21.7 ± 2.4. Among the participants, 159 (19.9%) were first-, 154 (19.3%) second-, 112 (14.1%) third-, 117 (14.7%) fourth-, 117 (14.7%) fifth-, and 138 (17.3%) sixth- year dental students. According to the place of residence, 311 (39.0%) of the participants live with parents, 366 (45.9%) in university dormitories, and 120 (15.1%) in rented or owned apartments.

### 2.2. Instruments

The measurements used in the study were the Demographic and Academic Questionnaire, Questionnaire about exposure to COVID-19, COV19-impact on quality of life (COV19-QoL) questionnaire, Generalized Anxiety Disorder 7-item (GAD-7) scale, and Patient Health Questionnaire (PHQ-8).

The Demographic and Academic Questionnaire contains questions about age, sex, place of residence, and year of studies. The exposure to COVID-19 was measured by seven questions concerning pandemic consequences related to (1) being infected with COVID-19; (2) experiencing COVID-19 symptoms; (3) being tested for COVID-19; (4) being hospitalized due to COVID-19; (5) being in a strict quarantine; (6) COVID-19 infection in family, friends, or relatives; and (7) death in the family and close relatives [32].

#### 2.2.1. COV19-QoL Questionnaire

The COV19-QoL is a brief, unidimensional instrument that contains six questions on a 5-point Likert scale (1 = completely disagree, 2 = disagree, 3 = neither agree nor disagree, 4 = agree, 5 = completely agree) [33]. It examines the impact of COVID-19 on the quality of life regarding mental health and it was developed on a sample from Balkan general and clinical population. Questions explore the sense of impact on one’s quality of life, mental and physical health decline, anxiety, depression, and personal safety. The questionnaire was translated into Serbian and back-translated to ensure that the expressions were appropriate. The Cronbach’s alpha was 0.84, indicating a high level of internal consistency for this sample. This questionnaire provides acceptable psychometric characteristics and is of adequate quality [29].

#### 2.2.2. Generalized Anxiety Disorder 7-Item Scale (GAD-7)

The 7-item Generalized Anxiety Disorder (GAD-7) scale in the Serbian adaptation is a self-reported measure designed to screen for symptoms following the Diagnostic and Statistical Manual of Mental Disorders, fourth edition (DSM-IV) criteria [30,32,34]. Participants rate how often they experienced anxiety symptoms in the two weeks preceding the study on a 4-point Likert scale (0 = not at all, 1 = several days, 2 = more than half the days, and 3 = nearly every day). The ranges of GAD-7 scores are 0–4 minimal anxiety; 5–9 mild anxiety; 10–14 moderate anxiety; and 15–21 severe anxiety [35]. The validated cut-off score of ≥10 has been recommended as an indicator for moderate/severe symptoms of generalized anxiety disorder. The GAD-7 demonstrates adequate internal consistency (Cronbach’s alpha was 0.85).

#### 2.2.3. Patient Health Questionnaire (PHQ-9)

Symptoms of depression were assessed using the Patient Health Questionnaire (PHQ-9) questionnaire in the Serbian adaptation. This scale is widely used to assess symptoms of depression and represents a screening gold standard [31]. It has nine items scoring nine common symptoms of depression in the past two weeks. It has a 4-point rating scale (0 = not at all, 1 = several days, 2 = more than half the days, 3 = nearly every day). Scores indicate as follows: 5–9 mild; 10–14 moderate; 15–19 moderate to severe, and 20 and above severe depression. The validated cut-off score of ≥10 was recommended as an indicator for moderate to severe depression symptoms [36]. For the 9th question of PHQ-9 (questioning if there were any “thoughts that you would be better off dead or of hurting yourself in some way”), the cut-off score of ≥1 was used as an indicator of suicidality (endorsement of “several days” or more to the items). In this study, the Cronbach’s alpha for the PHQ-9 equaled 0.86.

The survey lasted around 25 min.

### 2.3. Statistical Analysis

Normality of a continuous distribution was assessed using skewness and kurtosis statistics (below an absolute value of 2.0). Several different methods were used: descriptive summary statistics for the demographic, academic characteristics, exposure to COVID-19, and COV19-QoL, GAD-7, and PHQ-9 scores; parametric (*t*-test) and non-parametric statis-tic tests (χ^2^ test, Fisher exact test, Mantel–Haenszel test for trend) to determine demographic and academic characteristics within the sample; Pearson’s coefficient in order to verify correlations among the variables; one-way analysis of variance (ANOVA) to evaluate the significance of differences; regressive multivariable analysis (linear regression) to identify the predictors of a better perception of QoL, symptoms of Generalized Anxiety Disorder, and symptoms of depression. Software package SPSS ver. 20 was used for the analyses (SPSS Inc., Chicago, IL, USA).

## 3. Results

The coronavirus-related and psychological variables are presented in Table 1. About half of the students were infected, had symptoms, and were tested for COVID, while 2.1% were hospitalized. Every second student was in strict quarantine for at least 14 days. Almost every third student reported a COVID-19 infection in close relatives, while every fourth experienced the death of a close relative caused by COVID-19.

Based on the analysis of COV19-impact on quality-of-life questionnaire (mean value and standard deviation were 2.9 ± 0.9), almost half of the participants think that their quality of life is lower and feel tenser than before the pandemic. More than a third of students think their mental and physical health may have deteriorated and felt more depressed than before the pandemic. The lowest number of respondents was concerned about their personal safety (Table 1). The highest total score was for the pandemic impact on quality of life and increased feelings of tension (3.29 ± 1.17, 3.28 ± 1.24, respectively), while the lowest score was for concerns about personal safety 2.35 ± 1.19.

According to the results of the analysis of GAD-7 and PHQ-9 questionnaire (mean value and standard deviation were 5.4 ± 5.1 and 7.6 ± 5.7, respectively), every fifth student showed the symptoms of general anxiety disorder (moderate, severe) and almost a third of students met the diagnostic criteria of depression (moderate, moderately severe, and severe). Every sixth student showed a dual anxiety and depression diagnosis (Table 1).

The relation between quality of life and other variables (anxiety and depression) was significantly correlated (Table 2). The effect size of the correlation between the quality of life and anxiety and depression was positive and high. General anxiety disorder positively correlated with depression disorder and the effect size was also strong.

There is a statistically significant difference in the distribution according to gender, age, and the presence of symptoms in relation to the year of study (Table 3). In all years of study, there are statistically significantly more female than male students. The highest number of male students is in the sixth year of study. Second- and sixth-year students had the most students with symptoms of COVID-19.

The presence of symptoms, testing, hospitalization, quarantine, and occurrence of illness in close relatives was statistically more frequent in students who were infected with COVID-19 (*p* ˂ 0.001) (Table 3). Out of the total number of students who had a COVID-19 infection, 8.5% were asymptomatic and had no symptoms.

We have performed ANOVA analysis to determine if there is any statistically significant difference in quality of life, general anxiety levels, and level of depression between different genders, places of residence, years of study, infection with COVID-19, and experiences with the death of close relatives due to the pandemic (Table 4). The results indicate that female students have statistically significant higher scores on QoL and GAD-7 than male students (*p* = 0.040, *p* = 0.009, respectively). Students living in dormitories have significantly higher scores for GAD-7 (*p* = 0.025), while students living with parents have significantly higher scores for PHQ-9 (*p* = 0.009) than others. There is a statistically significant difference in quality of life, anxiety, and depression in relation to the year of study. COVID-19 had a statistically significantly greater impact on second- and third-year students compared to first- (*p* = 0.009, *p* = 0.013, respectively) and fifth-year (*p* = 0.001, *p* = 0.001, respectively) students. The level of anxiety was statistically significantly higher among students in the second and third years compared to students in the first (*p* = 0.047), fifth (*p* = 0.001, *p* = 0.030, respectively) and sixth years of study (*p* = 0.001, *p* = 0.040, respectively). The level of depression was significantly more pronounced among students in the first (*p* = 0.004), second (*p* = 0.001), and third (*p* = 0.007) years compared to students in the fifth year of study. Additionally, students in the second year had more pronounced symptoms of depression compared to students in the sixth year of study. There was no statistically significant difference in the quality of life and presence of symptoms of anxiety and depression between students who had and those who did not have COVID-19. Students who report the death of a close relative due to coronavirus had much higher scores on the QoL scale total score compared to students who did not report it (*p* = 0.002).

Linear regression analysis identified significant demographic, academic, and exposure to COVID-19 predictors of lower perception of quality of life, higher level of anxiety, and depression among dental students during the COVID-19 pandemic (Table 5). The dependent variables introduced into the regression model were the following: perception of the negative impact of COVID-19 on well-being, anxiety, and depression. Gender and the death of a close relative due to coronavirus were predictors for the lower perception of quality of life during COVID-19. Place of residence, gender, and the death of a close relative were predictors for higher levels of generalized anxiety disorder, while the year of study, place of residence, and death of a close relative due to coronavirus were predictors for the onset of depression.

## 4. Discussion

A recent study showed that the COVID-19 pandemic and epidemiological measures to prevent the spread of infection significantly impact public mental health and quality of life [37]. Evidence from previous large-scale health outbreaks suggests that pandemics have a tremendous impact on young people, which current pandemic research confirms [14,38,39]. Many mental health problems have worsened in the student population during the COVID-19 pandemic [11,17,31].

The data from the previous pandemic wave, which preceded our research, showed that 27.3% of Belgrade University students had mild, 59.3% moderate, and 13.4% severe symptoms of COVID-19 infection [40]. In our research, only 2.1% of students were hospitalized, which is associated with a severe symptom. Similar results were obtained in other studies [28,32].

It is known from the literature that COVID-19 has a negative impact on the HRQoL of all population groups [5,41]. The total and particular scores of COV19-QoL were similar to a previous study conducted in the Balkan region among non-clinical patients. Our respondents were less concerned about personal safety, which is understandable, considering that they are younger than in the mentioned study [33].

In Greece, results showed a deterioration of the QoL during the lockdown in more than half of the university students and between 21% and 54% of nursing students [42,43]. It is noteworthy that COVID-19 impaired the QoL of university medical students even after the lockdown was lifted [44]. In our study, about half of the respondents reported a worse QoL during the pandemic than before the pandemic.

Some students might be at a higher risk for lower perceptions of quality of life during the COVID-19 pandemic. Studies before the pandemic showed that female medical students had lower QoL scores than male students [45]. Recent research among dental master and doctoral students indicated that female students had lower scores on QoL [46]. On the contrary, others did not recognize the impact of gender on QoL [47]. Another study reported a lower perception of overall well-being and quality of life in students who are more emphatic about their family and friends [13]. Similarly, our results suggest that the female gender and the death of close relatives were predictors for the lower perception of quality of life due to COVID-19.

Early findings in China have suggested that more than 28.8% of the general population experienced moderate to severe levels of anxiety-related symptoms in response to COVID-19 [39]. In Serbia, the prevalence of anxiety in the general population during lockdown was 36.9% [14]. Some studies reported that the anxiety prevalence among students is higher than in the general population, especially during the COVID-19 pandemic [12,13,32]. The prevalence of moderate and severe anxiety symptoms in university students ranges from 52.0% in Turkish, 42.5% in Greece, and 37.0% in Malaysian students [32,42,43,48]. The observed anxiety symptom levels in our study were lower than in the study conducted under the lockdown and strict epidemiological measures. Additionally, they were similar to the anxiety level of dental students from other university center [28]. However, we emphasized the fact that there was a great diversity in the questionnaire selection and survey conditions between studies.

The results of our study indicate that students living in the dormitory had higher anxiety than students living with their parents or on their own, which is in accordance with the literature [34]. Close contact between students in dormitories and various mutual activities, such as eating in the canteen, sport, and social events, may increase the likelihood of viral spread, which may explain the higher degree of anxiety [40,49].

Many studies showed that females are psychologically more affected by the COVID-19 pandemic than males [13,48,50]. In addition, several studies, including ours, have reported higher anxiety levels among female students during the COVID-19 crisis [13,32,51].

Depression is one of the most common mental health disorders characterized by numerous symptoms, including depressed mood, loss of interest in most/all activities, loss of energy, or feelings of worthlessness [52]. The findings present that young people were more often diagnosed with depression during the COVID-19 crisis [51,53,54]. The recent study found that 28.9% of the Serbian population had moderate to severe depression during the COVID-19 pandemic, with the highest prevalence among university students [14]. Our results show that 31.4% of students had depression symptoms. Interestingly, in Serbia, the prevalence of depression among dental students was lower (12.4%) [28]. An explanation of these results could be the use of another survey instrument as well as conducting research during the semester break.

In our study, students living with their parents had two times more pronounced symptoms of depression compared to students living in a dormitory. A possible explanation for these findings could be that students who live in a dormitory with their peers have more social contacts and better social support.

The negative psychological impact on people who are more worried about a family member or who experienced death in a close environment caused by COVID-19 has been found in our study and recognized by others [13,55]. We also found that a lower perception of quality of life, anxiety, and depression was significantly associated with the death of a close relative.

One study report that fourth-year students were more stressed due to a lack of clinical skills, not passing the clinic/skills courses due to the lack of study progression, and worries about not being a good enough dentists after graduation [22]. However, the results of our research show that the year of study is a predictor for higher depression level. Students in their second and third year of study had a higher level of QoL disorders, anxiety, and depression than students in the first, fifth, and sixth year of study. Since the clinical exercises start in this period, they could be the reason for obtaining previously mentioned results. First-year students still have only theoretical classes, while students in older years of study already have some experience, which makes them less concerned.

Among the students who reported being infected with COVID-19, there are significantly more of those who had symptoms, were tested, hospitalized, have been in quarantine, and have infected relatives or household members. We registered, that among students who had COVID-19, every twelfth was asymptomatic. It is known that asymptomatic carriers of COVID-19 may increase the transmission of infection. One meta-analysis reported more than 50% of asymptomatic cases among COVID-positive subjects [56]. Our percentage is significantly lower, but we assume that among those who had no symptoms and were not tested, there is a certain percentage of asymptomatic subjects.

In our research, there is no significant difference in the quality of life, anxiety, and depression between students who were infected with SARS-CoV-2 and those who did not. These results suggest that other variables significantly influenced students’ overall quality of life and mental health compared to the COVID-19 infection. Other studies concluded that social life, social support, and the manner of teaching had a more significant impact on the quality of life and mental health of students [7,13,20,24,25].

There are some limitations of the study that should be highlighted. Knowing that all subjects included in the study were students from the same university, a bigger sample size from multiple institutions could provide a better perspective. Furthermore, this is a cross-sectional study and addresses the present time only. Therefore, there is a need for a prospective study that will monitor mental health and quality of life over a prolonged period.

Consequently, dental students’ quality of life and mental health during the pandemic requires monitoring and in-depth research. Dental schools should identify and support students at a higher risk of negative psychological effects during unusual situations such as the COVID-19 pandemic.

## 5. Conclusions

This study showed that dental students’ mental health during the pandemic is at a high risk, especially in female students, second- and third-year students, students living in a dormitory or with parents, and students who had experienced a death in a close environment caused by COVID-19. However, infection with COVID-19 did not have a significant impact on the QoL and mental health of dental students. It is necessary to work on future strategies related to combining online teaching with on-site courses for future pandemics and emergencies. Additionally, universities should consider students’ psychological and mental health during the pandemic. Dental schools should establish psychological support programs with development techniques for overcoming crises such as a pandemic with cooperation with other health and educational institutions.

## Figures and Tables

**Table 1 ijerph-19-14061-t001:** Coronavirus-related and psychological variables (n = 797).

Variable	n	%
Exposure to COVID-19		
Infected with COVID-19	377	47.3
Symptoms of coronavirus infection	399	50.1
Tested for coronavirus	479	58.1
Hospitalization due to coronavirus	17	2.1
Strict quarantine for at least 14 days	368	46.2
Coronavirus infection in close relatives	556	69.8
Death of close relative due to coronavirus	194	24.3
Quality of life (Completely agree, Agree)		
1. I think my quality of life is lower than before	377	47.3
2. I think my mental health has deteriorated	285	35.7
3. I think my physical health may deteriorate	262	32.9
4. I feel more tense than before	401	50.3
5. I feel more depressed than before	278	34.9
6. I feel that my personal safety is at risk	137	17.2
Anxiety (GAD-7)		
Normal (0–4)	425	53.5
Mild (5–9)	211	26.6
Moderate (10–14)	92	11.5
Severe (15–21)	69	8.4
Depression (PHQ-9)		
Normal (0–4)	305	38.3
Mild (5–9)	242	30.3
Moderate (10–14)	129	17.5
Moderately severe (15–19)	81	10.2
Severe (20–24)	29	3.7
Neither depression nor anxiety diagnosis (score < 10)	517	64.9
Anxiety only diagnosis (GAD-7 > 10)	155	19.9
Depression only diagnosis (PHQ-9 > 10)	248	31.2
Dual anxiety and depression diagnosis (score > 10)	124	15.5

**Table 2 ijerph-19-14061-t002:** Correlation matrix between COV19-impact on quality of life and GAD-7 and PHQ-9 with Pearson’s r coefficient (n = 797).

Variable	COV19-Impact on Quality of Life	GAD-7
**GAD-7**	0.609 **	-
**PHQ-9**	0.613 **	0.750 **

Remark: statistically significant results correspond to ** *p* < 0.001.

**Table 3 ijerph-19-14061-t003:** Demographic, academic, and exposure to COVID-19 variables according to the year of study and infection with COVID-19 (n = 797).

Variable	Year of Study n (%)							Infected with COVID-19 n (%)		
	First	Second	Third	Fourth	Fifth	Sixth	*p*	Yes	No	*p*
**Gender**										
** Male**	38 (4.8)	34 (4.3)	27 (3.4)	31 (3.9)	32 (4.0)	46 (5.8)	0.041 *	110 (13.8)	97 (12.2)	0.051
** Female**	121 (15.2)	120 (15.1)	85 (10.7)	86 (10.8)	85 (10.7)	92 (11.3)		267 (33.5)	323 (40.5)	
**Average age**	18.9 ± 1.2	20.0 ± 1.5	21.3 ± 2.1	22.3 ± 2.6	23.7 ± 2.2	24.2 ± 2.7	<0.001 *	21.8 ± 2.5	21.6 ± 2.3	0.333
**Place of residence**										
** With parents**	62 (7.8)	52 (6.5)	55 (6.9)	44 (5.5)	46 (5.8)	52 (6.5)	0.993	149 (18.7)	162 (20.3)	0.092
** In university dormitories**	28 (3.5)	23 (2.9)	12 (1.5)	14 (1.8)	22 (2.8)	21 (2.6)		46 (5.8)	74 (9.3)	
** In rented or own apartments**	69 (8.7)	79 (9.9)	45 (5.6)	59 (7.4)	49 (6.1)	65 (8.2)		182 (22.8)	184 (23.1)	
**Exposure to COVID-19**										
**Infected with COVID-19**										
** Yes**	61 (7.7)	74 (9.3)	58 (7.3)	59 (7.4)	55 (6.9)	70 (8.8)	0.210			
** No**	98 (12.3)	80 (10.0)	54 (6.8)	58 (7.3)	62 (7.8)	68 (8.4)				
**Symptoms of coronavirus infection**										
** Yes**	65 (8.2)	75 (9.4)	62 (7.8)	62 (7.8)	62 (7.8)	73 (9.2)	0.033 *	345 (43.3)	54 (6.8)	˂0.001 *
** No**	94 (11.8)	79 (10.9)	50 (6.3)	55 (6.3)	55 (6.3)	65 (8.2)		32 (4.1)	366 (45.8)	
**Tested for coronavirus**										
** Yes**	82 (10.4)	96 (12.1)	73 (9.3)	68 (8.6)	74 (9.4)	86 (10.9)	0.123	328 (41.2)	151 (18.9)	˂0.001 *
** No**	77 (9.8)	58 (7.4)	39 (4.9)	49 (6.2)	43 (4.4)	52 (6.6)		49 (6.1)	269 (33.8)	
**Hospitalization due to coronavirus**										
** Yes**	7 (0.8)	3 (0.4)	2 (0.3)	2 (0.3)	1 (0.2)	2 (0.3)	0.067	17 (2.1)	0 (0.0)	˂0.001 *
** No**	152 (19.1)	151 (18.8)	110 (13.7)	115 (14.3)	116 (14.5)	136 (17.3)		360 (45.2)	420 (52.7)	
**Strict quarantine for at least 14 days**										
** Yes**	68 (8.5)	68 (8.5)	60 (7.5)	54 (6.8)	54 (6.8)	64 (8.0)	0.556	338 (42.4)	30 (3.8)	˂0.001 *
** No**	91 (11.5)	86 (10.8)	52 (6.5)	63 (7.9)	63 (7.9)	74 (9.3)		39 (4.9)	390 (48.9)	
**Coronavirus infection in close relatives**										
** Yes**	102 (12.7)	113 (14.2)	82 (10.3)	83 (10.4)	81 (10.2)	95 (11.9)	0.709	323 (40.5)	233 (29.2)	˂0.001 *
** No**	57 (7.2)	41 (5.1)	30 (3.8)	34 (4.3)	36 (4.5)	43 (5.4)		54 (6.8)	187 (23.5)	
**Death of close relative due to coronavirus**										
** Yes**	36 (4.5)	45 (5.6)	33 (4.1)	24 (3.0)	26 (3.3)	30 (3.8)	0.300	100 (12.5)	94 (11.8)	0.178
** No**	123 (15.4)	109 (13.8)	79 (9.9)	93 (11.7)	91 (11.4)	108 (13.7)		277 (34.8)	326 (40.9)	

Remark: statistically significant results correspond to * *p* < 0.05.

**Table 4 ijerph-19-14061-t004:** The impact of gender, place of residence, years of study, infection with COVID-19, and death of close relatives on total scores of COV19-impact on quality of life, GAD-7, and PHQ-9 (n = 797).

	COV19-Impact on Quality of Life			GAD-7			PHQ-9		
	Mean (SD)	F	*p*	Mean (SD)	F	*p*	Mean (SD)	F	*p*
**Gender**									
** Male**	2.7 ± 0.9	4.3	0.040 *	4.6 ± 4.9	6.9	0.009 *	7.0 ± 5.6	2.8	0.093
** Female**	2.9 ± 0.9			5.7 ± 5.2			7.8 ± 5.9		
**Place of residence**									
** With parents**	3.0 ± 0.5	1.1	0.327	5.9 ± 5.4	3.7	0.025 *	8.1 ± 6.0	4.8	0.009 *
** In dormitories**	2.9 ± 0.6			6.2 ± 5.0			4.4 ± 0.4		
** In rented or own apartments**	2.9 ± 0.9			5.4 ± 5.0			7.6 ± 5.5		
**Year of studies**									
** First**	2.7 ± 0.9	5.9	<0.001 *	5.3 ± 4.5	7.6	0.003	8.1 ± 5.4	5.7	0.019
** Second**	3.1 ± 0.9			6.9 ± 5.6			9.3 ± 5.9		
** Third**	3.1 ± 0.9			6.3 ± 5.4			8.3 ± 5.8		
** Fourth**	2.9 ± 0.9			5.4 ± 5.4			7.4 ± 5.8		
** Fifth**	2.7 ± 0.9			4.2 ± 4.3			5.7 ± 4.9		
** Sixth**	2.9 ± 0.9			4.4 ± 4.8			6.3 ± 5.3		
**Infected with COVID-19**									
** Yes**	3.0 ± 0.9	2.8	0.093	5.4 ± 5.0	0.1	0.943	7.7± 5.7	0.2	0.669
** No**	2.9 ± 0.9			5.5 ± 5.2			7.5 ± 5.6		
**Death of close relatives**									
** Yes**	3.1 ± 0.8	10.1	0.002 *	6.3 ± 5.3	6.7	0.010 *	8.5 ± 5.7	7.1	0.008 *
** No**	2.8 ± 0.8			5.2 ± 5.0			7.3 ± 5.6		

Remark: statistically significant results correspond to * *p* < 0.05.

**Table 5 ijerph-19-14061-t005:** Results of linear regression to identify sociodemographic and exposure to COVID-19 predictors of COV19-impact on quality of life, GAD-7, and PHQ-9 score from dental students (n = 797).

	COV19-Impact on Quality of Life				GAD-7				PHQ-9			
	Unstandardized Coefficients				Unstandardized Coefficients				Unstandardized Coefficients			
	B	Std. Error	*t*	*p*	B	Std. Error	*t*	*p*	B	Std. Error	*t*	*p*
(Constant)	3.960	0.68	5.84	<0.001 *	8.686	3.80	2.29	0.023 *	11.698	4.18	3.28	0.001 *
Gender	0.15	0.73	1.97	0.049 *	0.97	0.41	2.34	0.019 *	0.56	0.45	1.23	0.216
Age	−0.02	0.03	−0.81	0.421	0.01	0.14	0.48	0.962	−0.01	0.16	−0.05	0.957
Year of study	0.00	0.04	0.07	0.942	−0.34	0.19	−1.74	0.082	−0.54	0.21	−2.53	0.012 *
Place of residence	−0.05	0.05	−1.13	0.266	−0.66	0.26	−2.54	0.011 *	−0.76	0.29	−2.66	0.008 *
Infected with COVID-19	0.03	0.15	1.18	0.860	0.47	0.83	0.57	0.572	1.09	0.91	1.2	0.230
Symptoms of coronavirus infection	0.05	0.10	0.48	0.630	0.86	0.58	0.15	0.883	−2.84	0.64	−0.44	0.658
Tested for coronavirus	−0.04	0.08	−0.51	0.609	−0.32	0.43	−0.55	0.454	−0.75	0.48	−1.57	0.117
Hospitalization due to coronavirus	−0.06	0.23	−0.26	0.798	−0.61	1.27	−0.48	0.633	−0.55	1.39	−0.39	0.691
Strict quarantine for at least 14 days	−0.15	0.12	−1.31	0.190	−0.65	0.65	−0.99	0.322	−0.97	0.72	−1.36	0.174
Coronavirus infection in close relatives	−0.08	0.07	−1.087	0.283	0.52	0.42	1.23	0.218	0.34	0.46	0.73	0.466
Death of close relative due to coronavirus	−0.21	0.07	−2.08	0.005 *	−0.93	0.42	−2.21	0.028 *	−1.01	0.46	−2.19	0.029 *

Remark: statistically significant results correspond to * *p* < 0.05.

## Data Availability

The datasets generated during and/or analyzed during the current study are available from the corresponding author upon reasonable request.

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
