# Peer review of "Mental Health and Quality of Life among Dental Students during COVID-19 Pandemic: A Cross-Sectional Study"

_ijerph, 2022, doi:10.3390/ijerph192114061_

Round 1

Reviewer 1 Report

This is an interesting paper with valuable insight into a much-needed area of research.   Abstract The abstract introduces the content of the paper appropriately.    Introduction The introduced theories and previous research results are in line with the content and direction of the paper. However, besides the aims (which are very brief), basic research questions or exact hypotheses should be introduced too, possible at the end of the Introduction section (after introducing the aims).   Material and methods The Participants subchapter is very confusing; it should be revised.
Line 62-68: not relevant in the Participants subchapter; it should be replaced at the beginning of the paper (Introduction) Line 68-75: it is not clear whether th​e​ author is talking about the current research or generally, it should be specified, and the information should be kept here only if it is the introduction of the current research (however, there are some information in the next paragraph as well which is confusing)
The content of the Instrument subchapter is appropriate, the applied tools are well-interpreted. It would be great to​ read about the Cronbach alpha score of the GAD-7 Concerning the Statistical analysis subchapter, what was the point of choosing parametric tests and non-parametric ones? Did the authors use normality tests? Normality tests can show us the distribution of the data, which can help us in the choice of tests.   Results ​Results are also clear.  ​Line 146-151 should be placed ​​in​ the Participants subchapter   ​Discussion ​Discussion is correctly defined. However, it should be necessary to reflect on the assumptions of the authors, i.e. the research questions of hypotheses. This is now missing; however, research findings are compared to previous research results in a correct manner. Line 324-238 is evidence, it can be deleted​ ​Line 244-247: these findings ​should be compared to the results of the current research   ​Conclusions It would be important to highlight the possibilities and practices for developing  the mental health of students ​General comments ​Table4: in the Remark (below the table), P should be changed to p English language should be revised due to the small grammatical mistakes and typos. ​ ​In my opinion, this paper is an intriguing paper introducing extremely important practical information. It is worth publishing this paper after modifying the content mentioned above.​

Author Response

Response to Reviewer 1 Comments

Dear reviewer,

Thank you for your review, your useful suggestions and remarks. We are grateful for the opportunity to respond, and here we attach the answers to the specific comments.

Point 1:  The introduced theories and previous research results are in line with the content and direction of the paper. However, besides the aims (which are very brief), basic research questions or exact hypotheses should be introduced too, possible at the end of the Introduction section (after introducing the aims).

Response 1: We adopted the suggestion and corrected text in Introduction section:

The aims of the present study were to (1) investigate the impact of the COVID-19 pandemic on the quality of life and mental health (anxiety and depression) and (2) identify significant predictors of the quality of life, levels of anxiety and depression in a sample of dental students by analyzing a number of demographic and academic characteristics and exposure to COVID-19.

The findings identify a vulnerable subpopulation of dental students who should get special attention in order to preserve and improve their quality of life and mental health.

Point 2:  The Participants subchapter is very confusing; it should be revised. Line 62-68: not relevant in the Participants subchapter; it should be replaced at the beginning of the paper (Introduction). Line 68-75: it is not clear whether th​e​ author is talking about the current research or generally, it should be specified, and the information should be kept here only if it is the introduction of the current research (however, there are some information in the next paragraph as well which is confusing).

Response 2: We adopted the suggestion and corrected text in Material and Methods (Participants subchapter) reads:

Participants

A cross-sectional study was conducted at the begging of the winter semester 2021 (between 4th-8th October) during the mandatory introductory practical classes in classrooms at the School of the Dental Medicine University of Belgrade, Serbia. The students were selected in the order of appearance regardless of the year of study. Participation was anonymous and voluntary. All students provided written informed consent to participate in this study. 

The sample consisted of 867 students. Forty-three students refused to participate in the study, while 27 provided invalid data. The final sample included 797 students, male (n=207, 26%) and female (n=592, 74%) with average age 21.7 ± 2.4. Among participants, 159 (19.9%) were the 1st, 154 (19.3%) the 2nd, 112 (14.1%) the 3rd, 117 (14.7%) the 4th, 117 (14.7%) the 5th and 138 (17.3%) the 6th year dental students. According to the place of residence, 311 (39.0%) of participants live with parents, 366 (45.9%) in university dormitories, and 120 (15.1%) in rented or owned apartments.

Point 3: It would be great to​ read about the Cronbach alpha score of the GAD-7.

Response 3: We added Cronbach alpha score of the GAD-7 in Material and Methods (Instruments subchapter):

The GAD-7 demonstrate adequate internal consistency (Cronbach’s alpha was 0.85).

Point 4: Concerning the Statistical analysis subchapter, what was the point of choosing parametric tests and non-parametric ones? Did the authors use normality tests? Normality tests can show us the distribution of the data, which can help us in the choice of tests.  

Response 4: Indeed, we did not express ourselves well. We use non-parametric tests for checking the normality of distribution of the data, on the basis of which we further selected the tests. The text has been amended to read:

Normality of a continuous distribution was assessed using skewness and kurtosis statistics (below an absolute value of 2.0). Several different methods were used: descriptive summary statistics for the demographic, academic characteristics, exposure to COVID-19 and COV19-QoL, GAD-7 and PHQ-9 scores; parametric (t-test) and non-parametric statistic tests (χ2 test, Fisher exact test, Mantel-Haenszel test for trend) to determine demographic and academic characteristics within the sample; Pearson’s coefficient in order to verify correlations among variables; one-way analysis of variance (ANOVA) to evaluate significance of differences; regressive multivariable analysis (linear regression) to identify the predictors of a better perception of QoL, symptoms of Generalized Anxiety Disorder and symptoms of depression. Software package SPSS ver. 20 was used for the analyses (SPSS Inc, Chicago, USA).

Point 5: Line 146-151 should be placed in the Participants subchapter.

Response 5: We adopted the suggestion and that paragraph was added to the Participants subchapter:

The sample consisted of 867 students. Forty-three students refused to participate in the study, while 27 provided invalid data. The final sample included 797 students, male (n=207, 26%) and female (n=592, 74%) with average age 21.7 ± 2.4. Among participants, 159 (19.9%) were the 1st, 154 (19.3%) the 2nd, 112 (14.1%) the 3rd, 117 (14.7%) the 4th, 117 (14.7%) the 5th and 138 (17.3%) the 6th year dental students. According to the place of residence, 311 (39.0%) of participants live with parents, 366 (45.9%) in university dormitories, and 120 (15.1%) in rented or owned apartments.

Point 6: Discussion is correctly defined. However, it should be necessary to reflect on the assumptions of the authors, i.e. the research questions of hypotheses. This is now missing; however, research findings are compared to previous research results in a correct manner.

Response 6: The objectives of the research are reviewed in the Introduction chapter. In accordance with the changed objectives, the Discussion and Conclusion have been modified.

Point 7: Line 324-238 is evidence; it can be deleted.

Response 7: We adopt the recommendation. Sentences from the line have been deleted.

Point 8: Lie 244-247: these findings should be compared to the results of the current research.

Response 8: A sentence has been added:

In Greece, results showed deterioration of the QoL during the lockdown in more than half of university students and between 21% and 54% of nursing students [42,43]. It is noteworthy that COVID-19 impaired the QoL of university medical students even after the lockdown was lifted [44]. In our study, about half of the respondents reported a worse QoL during the pandemic than before the pandemic.

Point 9: Conclusions: It would be important to highlight the possibilities and practices for developing the mental health of students.

Response 9: We adopt the recommendation.

This study showed that dental students’ mental health during the pandemic is at high risk, especially in female students, 2nd and 3rd-year students, students living in a dormitory or with parents, and students who had experienced death in a close environment caused by COVID-19. However, the presence of the coronavirus symptoms did not have a significant impact on the QoL and mental health of dental students. It is necessary to work on future strategies related to combining online teaching with on-site courses for future pandemics and emergencies. Also, universities should consider students’ psychological and mental health during the pandemic. Dental schools should establish psychological support programs for development techniques for overcoming crises such as a pandemic with cooperation with other health and educational institutions.

Point 10: Table 4: in the Remark (below the table), P should be changed to p.

Response 10: Thank you for pointing out the omission. We replaced P with p.

Point  11: English language should be revised due to the small grammatical mistakes and typos.

Response 11: We revised the text with the help of an English language expert.

Reviewer 2 Report

1. Please elaborate the introduction, Kindly add dental studies if done in the same arena.

2. English grammar please check line 37 "many studies have identefied tremendous impact not only on physical health". Throughout the manuscript there is a need for moderate English correction.

3. Need of the study is unclear  so the objectives don't hold much options 

4.  Please start the study with " The study was approved by the Ethical Committee of the School of Dental Medicine, 84 University of Belgrade (No. 36/4)"  would suggest the authors to stick to STROBE guidelines for cross-sectional studies or what applicable. Kindly reframe

5. I suggest the authors to stratify the findings on yearly basis. Example: the Basic characteristics of students in 1 year , 2nd year and so on, and same applies to the scale and you can compare it suing RANOVA. 

6.  Do tell about the instruments in Intro

7. Table 1: Appears superfluous 

8. Table 2: Did you do the sub-analysis of participants, who were infected with COVID-19 (n=17)?? if not why 

9. Tested for coronavirus means those who tested positive ?? if yes then these variables should be analyzed separately to know about the actual impact , the findings should be compared with people who are not positive 

10. Discussion should change as there will be change in the test variables 

11. Conclusion should also be framed accordingly 

Author Response

Response to Reviewer 2 Comments

Dear reviewer,

Thank you for your review, your useful suggestions and remarks. We are grateful for the opportunity to respond, and here we attach the answers to the specific comments.

Point 1:  Please elaborate the introduction, Kindly add dental studies if done in the same arena.

Response 1: We adopted the suggestion and corrected text in Introduction section.

The Coronavirus Disease 2019 (COVID-19) was defined as extreme global health, economic and social emergency by the World Health Organization (WHO) on March 2020 [1]. This pandemic is a unique worldwide experience in modern history. Many studies identified its tremendous impact on physical health but also mental health and quality of life in general [2-5].

The first confirmed case of COVID-19 infection in Serbia was reported on 6th March 2020, and the first COVID-19-related death was announced on 2nd February. In Serbia, the lockdown has been implemented from March until May to prevent the spread of the infectious pandemic [6]. In that period, in addition to the suspension of activities at the university, all student dormitories were also closed. After reopening in May, an increasing number of students showed signs of infection by COVID-19 in several distinctive waves [7].

The University of Belgrade, as a public university located in a large metropolitan area, started using a hybrid teaching system. University organized online lectures with practical classes held physically on site, in reduced groups of students. All epidemiological recommendations given by the Ministry of Education, Science and Technological Development of Serbia were adopted to prevent the spread of COVID-19. In the following period, further enforcing social distancing, reducing working hours in stores and restaurants, gyms, sports facilities, and theaters, and limiting traveling and socializing were recommended. Studies show that epidemiological measures such as lockdowns and social isolation can decrease the spreading of Coronavirus but can also cause psychological distress, anxiety, and depression [4,8,9]. Although young people are less exposed, some studies have shown that epidemiological measures taken to prevent the spread of COVID infection had a more emotional effect on young people than on other age groups [10,11].

Regardless of good physical health, mental health disorder symptoms were widespread in the student population, exposing students as a particularly vulnerable group in terms of mental health during the COVID-19 pandemic [7,12-15]. Reports show that medical students, and particularly dental students, even before the pandemic, present a higher level of stress during education, which considerably impacts their quality of life (QoL) [7, 16,17].

Undergraduate programs in dentistry are characterized by extensive theoretical learning during the pre-clinical period, while the later period includes basic concepts for dental practice and the development of clinical skills necessary for professional activity. [16,17,18]. In both cases, the transfer of knowledge is more efficient in direct contact with teachers and patients and vital in acquiring clinical skills, which was hard to do during the pandemic. [19,20].

It is already known in the literature that the COVID-19 pandemic affected clinical dental education and clinical dental practice in general [21]. Furthermore, reports show that dental students are stressed due to a lack of clinical skills caused by the pandemic and worried about not becoming good enough dentists after graduation [22]. Due to the COVID-19 pandemic, there is a decrease in the quality of life among dental students who received online/distance learning [20]. Recent studies have shown that the population of dental students experienced significant levels of anxiety and depression during COVID-19. [20,23-28].

Psychometric scales are one of the most often used research methods in sciences providing reliable and valid measures of mental health status [29]. There are widely available psychometric tools developed to study the psychosocial impact of the COVID-19 pandemic, such as COV19 – impact on quality of life, GAD-7, and PHQ-9. These instruments presented good psychometric characteristics and quality in the general population [29-31].

The aims of the present study were to (1) investigate the impact of the COVID-19 pandemic on the quality of life and mental health (anxiety and depression) and (2) identify significant predictors of the quality of life, levels of anxiety and depression in a sample of dental students by analyzing a number of demographic and academic characteristics and exposure to COVID-19.

The findings identify a vulnerable subpopulation of dental students who should get special attention in order to preserve and improve their quality of life and mental health

Point 2: English grammar please check line 37 "many studies have identefied tremendous impact not only on physical health". Throughout the manuscript there is a need for moderate English correction.

Response 2: We revised the text with the help of an English language expert.

Point 3: Need of the study is unclear so the objectives don't hold much options. 

Response 3: We adopted the suggestion and corrected text in Introduction section:

The aims of the present study were to (1) investigate the impact of the COVID-19 pandemic on the quality of life and mental health (anxiety and depression) and (2) identify significant predictors of the quality of life, levels of anxiety and depression in a sample of dental students by analyzing a number of demographic and academic characteristics and exposure to COVID-19.

The findings identify a vulnerable subpopulation of dental students who should get special attention in order to preserve and improve their quality of life and mental health.

Point 4: Please start the study with " The study was approved by the Ethical Committee of the School of Dental Medicine, 84 University of Belgrade (No. 36/4)" would suggest the authors to stick to STROBE guidelines for cross-sectional studies or what applicable. Kindly reframe.

Response 4: We adopted the suggestion.

  1. Materials and Methods

The study was approved by the Ethical Committee of the School of Dental Medicine, University of Belgrade (No. 36/4) and conducted in accordance with the Declaration of Helsinki.

Participants

A cross-sectional study was conducted at the begging of the winter semester 2021 (between 4th-8th October) during the mandatory introductory practical classes in classrooms at the School of the Dental Medicine University of Belgrade, Serbia. The students were selected in the order of appearance regardless of the year of study. Participation was anonymous and voluntary. All students provided written informed consent to participate in this study. 

The sample consisted of 867 students. Forty-three students refused to participate in the study, while 27 provided invalid data. The final sample included 797 students, male (n=207, 26%) and female (n=592, 74%) with average age 21.7 ± 2.4. Among participants, 159 (19.9%) were the 1st, 154 (19.3%) the 2nd, 112 (14.1%) the 3rd, 117 (14.7%) the 4th, 117 (14.7%) the 5th and 138 (17.3%) the 6th year dental students. According to the place of residence, 311 (39.0%) of participants live with parents, 366 (45.9%) in university dormitories, and 120 (15.1%) in rented or owned apartments.

Point 5: I suggest the authors to stratify the findings on yearly basis. Example: the Basic characteristics of students in 1 year, 2nd year and so on, and same applies to the scale and you can compare it suing RANOVA. 

Response 5: We adopted the suggestion.

Table 4. Demographic, academic, and exposure to COVID-19 variables according to the year of study (n = 797).

Variable

Year of study

1st

2nd

3rd

4th

5th

6th

p

Gender n (%)

Male

38 (23.9)

34 (22.1)

27 (24.1)

31 (26.5)

32 (27.4)

46 (33.4)

0.041*

Female

121 (76.1)

120 (77.9)

85 (75.9)

86 (73.5)

85 (72.6)

92 (66.6)

Average age

18.9 ± 1.2

20.0 ± 1.5

21.3 ± 2.1

22.3 ± 2.6

23.7 ± 2.2

24.2 ± 2.7

<0.001*

Place of residence n (%)

With parents

62 (39.0)

52 (33.8)

55 (49.1)

44 (37.6)

46 (39.3)

52 (37.7)

0.993

In university

dormitories

28 (17.6)

23 (14.9)

12 (10.7)

14 (12.0)

22 (18.8)

21 (15.2)

In rented or own apartments

69 (43.4)

79 (51.3)

45 (40.2)

59 (50.4)

49 (41.9)

65 (47.1)

Exposure to COVID-19 n (%)

Symptoms of coronavirus infection

Yes

65 (40.9)

75 (48.7)

62 (53.0)

62 (53.0)

62 (53.0)

73 (52.9)

0.033*

No

94 (59.1)

79 (51.3)

50 (44.6)

55 (47.0)

55 (47.0)

65 (47.1)

Tested for coronavirus

Yes

82 (51.6)

96 (62.3)

73 (65.2)

68 (58.1)

74 (63.2)

86 (62.3)

0.123

No

72 (48.4)

58 (37.7)

39 (34.8)

49 (41.9)

43 (36.8)

52 (37.7)

Hospitalization due to coronavirus

Yes

7 (4.4)

3 (1.9)

2 (1.8)

1 (1.7)

1 (0.9)

2 (1.4)

0.067

No

152 (95.6)

151 (98.1)

110 (98.2)

115 (98.3)

116 (99.1)

136 (98.6)

Strict quarantine for at least 14 days

Yes

68 (42.8)

68 (44.2)

60 (53.6)

54 (46.2)

54 (46.2)

64 (46.4)

0.556

No

91 (57.2)

86 (55.8)

52 (46.4)

63 (53.8)

63 (53.8)

74 (53.6)

Coronavirus infection in close relatives

Yes

102 (64.2)

113 (73.4)

82 (73.2)

83 (70.9)

81 (69,2)

95 (68.8)

0.709

No

57 (35.8)

41 (26.6)

30 (26.8)

34 (29.1)

36 (30.8)

43 (31.2)

Death of close relative due to coronavirus

Yes

36 (22.6)

45 (29.2)

33 (29.5)

24 (20.5)

26 (22.2)

30 (21.7)

0.300

No

123 (77.4)

109 (70.8)

79 (70.5)

93 (79.5)

91 (77.8)

108 (78.3)

Point 6: Do tell about the instruments in Introduction

Response 6: A sentences has been added:

Psychometric scales are one of the most often used research methods in sciences providing reliable and valid measures of mental health status [29]. There are widely available psychometric tools developed to study the psychosocial impact of the COVID-19 pandemic, such as COV19 – impact on quality of life, GAD-7, and PHQ-9. These instruments presented good psychometric characteristics and quality in the general population [29-31].

Point 7: Table 1: Appears superfluous.

Response 7:  We adopted the suggestion and the Table 1 has been deleted.

Point 8:  Table 2: Did you do the sub-analysis of participants, who were infected with COVID-19 (n=17)?? if not why?

Response 8:  We assume you mean hospitalized patients. The proposal was adopted: Out of the total number of hospitalized students (n = 17, 2.1%), the majority were female students (n = 14, 82.4%), average age 22.0 ± 1.1, 1st year of study (n = 7, 41.2%), who live independently in their own apartments (n = 9, 52.9%).

Point 9: Tested for coronavirus means those who tested positive?? If yes, then these variables should be analyzed separately to know about the actual impact, the findings should be compared with people who are not positive. 

Response 9:  No, because testing for Covid does not necessarily mean a positive test for COVID-19. We decided to test the variable "Symptoms of coronavirus infection" in order to determine if there is a difference between students who had symptoms and those who did not (Table 5).

Point 10: Discussion should change as there will be change in the test variables.

Response 10:  We adopted the suggestion and added the text:

  • In our research, only 2.1% of students were hospitalized, which is associated with a severe symptoms. Similar results were obtained in other studies [28,32].
  • One study report that 4th-year students were more stressed due to a lack of clinical skills, not passing the clinic/skills courses due to lack of study progression, and worried about not being good enough dentists after graduation [22]. However, the results of our research show that 2nd and 3rd-year students had a higher level of QoL disorders, anxiety, and depression than students in the 1st, 5th, and 6th year of study. Since the clinical exercises start in this period, they could be the reason for obtaining previously mentioned results. First-year students still have only theoretical classes, while students in older years of study already have some experience, which makes them less concerned.
  • In our research, there is no significant difference in the quality of life, anxiety, and depression between students that had and those that did not have symptoms of COVID-19. These results suggest that other variables significantly influenced students' overall quality of life and mental health compared to the COVID-19 infection. Other studies concluded that social life, social support and the manner of teaching had a significant impact on the quality of life and mental health of students [7,13,20,24,25].

Point 11: Conclusion should also be framed accordingly 

Response 11:  We adopt the recommendation.

This study showed that dental students’ mental health during the pandemic is at high risk, especially in female students, 2nd and 3rd-year students, students living in a dormitory or with parents, and students who had experienced death in a close environment caused by COVID-19. However, the presence of the coronavirus symptoms did not have a significant impact on the QoL and mental health of dental students. It is necessary to work on future strategies related to combining online teaching with on-site courses for future pandemics and emergencies. Also, universities should consider students’ psychological and mental health during the pandemic. Dental schools should establish psychological support programs for development techniques for overcoming crises such as a pandemic with cooperation with other health and educational institutions.

Round 2

Reviewer 2 Report

Point 8: Table 2: Did you do the sub-analysis of participants, who were infected with COVID-19 (n=17)?? if not why?

Response 8: We assume you mean hospitalized patients. The proposal was adopted: Out of the total number of hospitalized students (n = 17, 2.1%), the majority were female students (n = 14, 82.4%), average age 22.0 ± 1.1, 1st year of study (n = 7, 41.2%), who live independently in their own apartments (n = 9, 52.9%).

Point 9: Tested for coronavirus means those who tested positive?? If yes, then these variables should be analyzed separately to know about the actual impact, the findings should be compared with people who are not positive.

Response 9: No, because testing for Covid does not necessarily mean a positive test for COVID-19. We decided to test the variable "Symptoms of coronavirus infection" in order to determine if there is a difference between students who had symptoms and those who did not (Table 5).

Response in Point 8 is unclear to me 

Point 9 : Even the test is negative for COVID-19 , and if there ARE SYMPTOMS, it might be regarded as asymptomatic carriers . Asymptomatic cases of SARS-CoV-2 can be unknown carriers magnifying the transmission of COVID-19. (refer 

Asymptomatic SARS-CoV-2 Carriers: A Systematic Review and Meta-Analysis Front. Public Health, 20 January 2021Sec. Infectious Diseases – Surveillance, Prevention and Treatment
https://doi.org/10.3389/fpubh.2020.587374)

Author Response

Dear reviewer,

Thank you again for your review, your useful suggestions and remarks. We are grateful for the opportunity to respond, and here we attach the answers to the specific comments.

Point 8:  Table 2: Did you do the sub-analysis of participants, who were infected with COVID-19 (n=17)?? if not why?

Response 8:  We adopted the suggestion and corrected text:

  • in the chapter Metherials and Methods

The Demographic and Academic Questionnaire contains questions about age, sex, place of residence, and year of studies. Exposure to COVID-19 was measured by seven questions concerning pandemic consequences related to 1) being infected with COVID-19; 2) experiencing COVID-19 symptoms; 3) being tested for COVID-19; 4) being hospitalized due to COVID-19; 5) being in a strict quarantine; 6) COVID-19 infection in family, friends, or relatives; 7) and death in the family and close relatives [32].

  • in the chapter Results:

The presence of symptoms, testing, hospitalization, quarantine and occurrence of illness in close relatives was statistically more frequent in students who were infected with the Coronavirus (p ˂ 0.001) (Table 3). Out of the total number of students who had a Coronavirus infection, 8.5% were asymptomatic and had no symptoms.

Table 3. Demographic, academic, and exposure to COVID-19 variables according to the year of study and infection with COVID-19 (n = 797).

Variable

Year of study n (%)

Infected with COVID-19

n (%)

1st

2nd

3rd

4th

5th

6th

p

Yes

No

p

Gender

Male

38 (4.8)

34 (4.3)

27 (3.4)

31 (3.9)

32 (4.0)

46 (5.8)

0.041*

110 (13.8)

97 (12.2)

0.051

Female

121 (15.2)

120 (15.1)

85 (10.7)

86 (10.8)

85 (10.7)

92 (11.3)

267 (33.5)

323 (40.5)

Average age

18.9 ± 1.2

20.0 ± 1.5

21.3 ± 2.1

22.3 ± 2.6

23.7 ± 2.2

24.2 ± 2.7

<0.001*

21.8 ± 2.5

21.6 ± 2.3

0.333

Place of residence

With parents

62 (7.8)

52 (6.5)

55 (6.9)

44 (5.5)

46 (5.8)

52 (6.5)

0.993

149 (18.7)

162 (20.3)

0.092

In university

dormitories

28 (3.5)

23 (2.9)

12 (1.5)

14 (1.8)

22 (2.8)

21 (2.6)

46

(5.8)

74 (9.3)

In rented or own apartments

69 (8.7)

79 (9.9)

45 (5.6)

59 (7.4)

49 (6.1)

65 (8.2)

182 (22.8)

184 (23.1)

Exposure to COVID-19

Infected with COVID-19

Yes

61 (7.7)

74 (9.3)

58 (7.3)

59 (7.4)

55 (6.9)

70 (8.8)

0.210

No

98 (12.3)

80 (10.0)

54 (6.8)

58 (7.3)

62 (7.8)

68 (8.4)

Symptoms of coronavirus infection

Yes

65 (8.2)

75 (9.4)

62 (7.8)

62 (7.8)

62 (7.8)

73 (9.2)

0.033*

345 (43.3)

54 (6.8)

˂0.001*

No

94 (11.8)

79 (10.9)

50 (6.3)

55 (6.3)

55 (6.3)

65 (8.2)

32

(4.1)

366 (45.8)

Tested for coronavirus

Yes

82 (10.4)

96 (12.1)

73 (9.3)

68 (8.6)

74 (9.4)

86 (10.9)

0.123

328 (41.2)

151 (18.9)

˂0.001*

No

77 (9.8)

58 (7.4)

39 (4.9)

49 (6.2)

43 (4.4)

52 (6.6)

49

(6.1)

269 (33.8)

Hospitalization due to coronavirus

Yes

7

(0.8)

3

(0.4)

2

(0.3)

2

(0.3)

1

(0.2)

2

(0.3)

0.067

17

(2.1)

0

(0.0)

˂0.001*

No

152 (19.1)

151 (18.8)

110 (13.7)

115 (14.3)

116 (14.5)

136 (17.3)

360 (45.2)

420 (52.7)

Strict quarantine for at least 14 days

Yes

68 (8.5)

68 (8.5)

60 (7.5)

54 (6.8)

54 (6.8)

64 (8.0)

0.556

338 (42.4)

30 (3.8)

˂0.001*

No

91 (11.5)

86 (10.8)

52 (6.5)

63 (7.9)

63 (7.9)

74 (9.3)

39

(4.9)

390 (48.9)

Coronavirus infection in close relatives

Yes

102 (12.7)

113 (14.2)

82 (10.3)

83 (10.4)

81 (10.2)

95 (11.9)

0.709

323 (40.5)

233 (29.2)

˂0.001*

No

57 (7.2)

41 (5.1)

30 (3.8)

34 (4.3)

36 (4.5)

43 (5.4)

54

(6.8)

187 (23.5)

Death of close relative due to coronavirus

Yes

36 (4.5)

45 (5.6)

33 (4.1)

24 (3.0)

26 (3.3)

30 (3.8)

0.300

100 (12.5)

94 (11.8)

0.178

No

123 (15.4)

109 (13.8)

79 (9.9)

93 (11.7)

91 (11.4)

108 (13.7)

277 (34.8)

326 (40.9)

  • in the chapter Discusion:

Among the students who reported being infected with the Coronavirus, there are sig-nificantly more of those who had symptoms, were tested, hospitalized, in quarantine and have infected relatives or household members. We registered that among students who had COVID-19 every twelfth was asymptomatic. It is known that asymptomatic carriers of COVID-19 may increase the transmission of infection. One meta-analysis reported more than 50% of asymptomatic cases among COVID positive subjects [56]. Our percentage is significantly lower, but we assume that among those who had no symptoms and who were not tested, there is a certain percentage of asymptomatic subjects.

In our research, there is no significant difference in the quality of life, anxiety, and depression between students who were infected with SARS-CoV-2 and those who did not. These results suggest that other variables significantly influenced students' overall quality of life and mental health compared to the COVID-19 infection. Other studies concluded that social life, social support and the manner of teaching had a more significant impact on the quality of life and mental health of students [7,13,20,24,25].

Point 9: Tested for coronavirus means those who tested positive?? If yes, then these variables should be analyzed separately to know about the actual impact, the findings should be compared with people who are not positive. 

Even the test is negative for COVID-19, and if there ARE SYMPTOMS, it might be regarded as asymptomatic carriers. Asymptomatic cases of SARS-CoV-2 can be unknown carriers magnifying the transmission of COVID-19 (https://doi.org/10.3389/fpubh.2020.587374).

Response 9:  We adopted the suggestion and corrected text:

  • in the chapter Results:

The presence of symptoms, testing, hospitalization, quarantine and occurrence of illness in close relatives was statistically more frequent in students who were infected with the Coronavirus (p ˂ 0.001) (Table 3). Out of the total number of students who had a Coronavirus infection, 8.5% were asymptomatic and had no symptoms.

  • in the chapter Discusion:

Among the students who reported being infected with the Coronavirus, there are significantly more of those who had symptoms, were tested, hospitalized, in quarantine and have infected relatives or household members. We registered that among students who had COVID-19 every twelfth was asymptomatic. It is known that asymptomatic carriers of COVID-19 may increase the transmission of infection. One meta-analysis reported more than 50% of asymptomatic cases among COVID positive subjects [56]. Our percentage is significantly lower, but we assume that among those who had no symptoms and who were not tested, there is a certain percentage of asymptomatic subjects.

In our research, there is no significant difference in the quality of life, anxiety, and depression between students who were infected with SARS-CoV-2 and those who did not. These results suggest that other variables significantly influenced students' overall quality of life and mental health compared to the COVID-19 infection. Other studies concluded that social life, social support and the manner of teaching had a more significant impact on the quality of life and mental health of students [7,13,20,24,25].
